# Life Cycle Data Interoperability Improvements through Implementation of the Federal LCA Commons Elementary Flow List

**Ashley N. Edelen** [1,*]**, Sarah Cashman** [1]**, Ben Young** [1] **and Wesley W. Ingwersen** [2]

1   Eastern Research Group, Inc., Lexington, MA 02421, USA
2   Center for Environmental Solutions and Emergency Response, Office of Research and Development,
    United States Environmental Protection Agency, Cincinnati, OH 45220, USA
*   Correspondence: ashley.edelen@erg.com

**Abstract:** As a fundamental component of data for life cycle assessment models, elementary flows have been demonstrated to be a key requirement of life cycle assessment data interoperability. However, existing elementary flow lists have been found to lack sufficient structure to enable improved interoperability between life cycle data sources. The Federal Life Cycle Assessment Commons Elementary Flow List provides a novel framework and structure for elementary flows, but the actual improvement this list provides to the interoperability of life cycle data has not been tested. The interoperability of ten elementary flow lists, two life cycle assessment databases, three life cycle impact assessment methods, and five life cycle assessment software sources is assessed with and without use of the Federal Life Cycle Assessment Commons Elementary Flow List as an intermediary in flow mapping. This analysis showed that only 25% of comparisons between these sources resulted in greater than 50% of flows being capable of automatic name-to-name matching between lists. This indicates that there is a low level of interoperability when using sources with their original elementary flow nomenclature, and elementary flow mapping is required to use these sources in combination. The mapping capabilities of the Federal Life Cycle Assessment Commons Elementary Flow List to sources were reviewed and revealed a notable increase in name-to-name matches. Overall, this novel framework is found to increase life cycle data source interoperability.

**Keywords:** nomenclature; life cycle assessment; elementary flow; interoperability





## 1. Introduction

Increasingly, life cycle assessment (LCA) is being used as an environmental management tool to make key decisions in reducing environmental impacts and optimizing product systems. LCA as a decision support tool is very data intensive, and most models utilize data from multiple data sources to provide the necessary background data or life cycle inventory (LCI) data to accompany the primary data used to model a product's life cycle [1]. The lack of harmonization across different data providers, software providers and life cycle impact assessment (LCIA) methods is a well-documented challenge [2–8]. Previous research efforts have been investigated to highlight and showcase the myriad of challenges associated with LCA data interoperability [9], even going so far as to create methodologies to review and assess the level of harmonization across various platforms [9,10]. Past research has highlighted the issues around nomenclature [7,10] and the need for improved semantics within LCA to access machine capabilities for harmonizing data across platforms [4,9,10]. These improvements are particularly important for elementary flows (EF), which are the foundational component of LCAs and link LCI and LCIA data to generate results.

Efforts on a global and U.S. national scale to harmonize LCA data have been ongoing since the early 2000s [11]. Global efforts for the harmonization of data in LCA are driven by the Life Cycle Initiative of the United Nations Environment Programme (UNEP) through

the Global Life Cycle Data Access (GLAD) initiative [12]. This initiative supported a global effort to analyze interoperability issues across LCA software, LCI databases and LCIA methods, and published the findings of these analyses in a critical review published in 2018 that produced 11 recommendations [10]. The evaluation of EF interoperability has been further improved with the methodology to develop pairwise mapping between major elementary flow lists developed through a project of the GLAD nomenclature working group [13].

On a national level, U.S. federal agencies have worked collaboratively through the Technical Working Group (TWG) on Federal LCA Data Interoperability initiated in 2014 to harmonize data across U.S. national agencies to produce a common data portal called the Federal LCA Commons [14–16]. As a part of this effort, the Federal LCA Commons decided to create a harmonized nomenclature of EFs called the Federal Elementary Flow List (FEDEFL). The objective of this paper is to analyze the ability of the adoption of the novel FEDEFL, built from the critical review recommendations [10], to effectively improve interoperability for LCA software, databases and LCIA method providers. The authors provide key elements of the FEDEFL framework in Section 2 to highlight the key components of the FEDEFL that were part of the analysis.

## 2. State of the Art of the Federal Elementary Flow List

This section highlights key components of the FEDEFL framework that are essential to the analysis. Detailed information on the FEDEFL structure can be found in a U.S. Environmental Protection Agency (USEPA) report [1]. The FEDEFL consists of a component-based structure, organizing and separating different data and metadata. The importance of this structure is to allow for increased automation in the handling of the list.

### 2.1. Federal Elementary Flow List Components

The creation of the FEDEFL was motivated by the need to be able to combine the individual components of a flow and eliminate the inconsistent inclusion of information in flows. There are two main types of components, the components of a flow and the metadata components, which are defined below.

#### 2.1.1. Elementary Flow Components

As defined by the FEDEFL USEPA report [1], EFs must have three components to identify them, which are required for a flow to appear in the FEDEFL:

1.  Flowable—The name of the material, energy, or space (e.g., "Carbon dioxide" or "freshwater") that comes from or goes to the biosphere.
2.  Context—A set of environmental media/compartments that describe the flow origin or destination.
3.  Unit—Flow units may be associated with conversion factors that can be used to convert between different units within a flow property (e.g., kg to lbs.) or even between flow properties (e.g., kg to $m^3$).

#### 2.1.2. Flow Metadata

Flow metadata consist of several components, including clarifiers, identifiers, unit converters, and secondary context information. Table 1 defines and provides examples of each component.

**Table 1.** Components of flow metadata.

| Metadata Component | Definition | Example(s) |
|---|---|---|
| Clarifiers | | |
| CAS Number | A unique numerical identifier assigned by the Chemical Abstracts Service | 10102-44-0 |
| Formula | Identifies each constituent element by its chemical symbol and indicates the proportionate number of atoms of each element | $NO_2$ |
| Synonyms | Another name for the same flowable | Nitrogen oxide |
| External Reference | External definition | Information on the substance details from USEPA's Substance Registry Services (SRS) |
| Identifiers | | |
| Flow UUID | A universally unique identifier to identify a flow (flowable + context) | b285ccbd-7703-39d9-9f66-20451f18d99f |
| Unit Converter | | |
| Alternate Unit | A unit other than the default unit for a flowable | kg |
| Conversion Factor | An arithmetical multiplier for converting from default to alternative units | 37 kg/MJ |

Defining the structure and the inclusion of metadata with flowables helps to eliminate metadata being mixed within the flow names. When EF lists mix metadata and flowables, the names become complex and automatic matching between flowables decreases significantly. This simplifying of the structure by defining the metadata components as separate from the flowable allows for increased automatic matching.

*2.2. Flow List Structure*
Flow Classes

The GLAD critical review developed flow classes as an additional flow metadata component, and these flow classes are used to organize flows and create common naming structures by flow class [10]. These flow classes were further refined in the FEDEFL USEPA report [1]. Flow classes are a way to group EFs by their potential uses in LCA data. Classes may have sets of contexts and units that distinguish them from flowables in other classes. Table 2 is the modified version of the flow classes used in the FEDEFL USEPA report [1].

**Table 2.** Flow classes.

| Flow Class | Resource/Emission | Definition | Example Flowable(s) |
|---|---|---|---|
| Biological | Both | Biomass or organic matter (i.e., microorganisms) | 'Wood' or 'Bacillus subtilis' |
| Element or Compound | Both | A unique chemical element or compound | 1,1,2,2-Tetrachloroethane |
| Energy | Both | Energy input NOT associated with consumed materials including heat | Energy, Geothermal |
| Geological | Both | A mineral or metal in an ore or aggregate material extracted for use or refining | Anthracite |
| Groups of Chemicals | Both | A group or mixture of chemicals | Dioxins |
| Land Use | Input | Land types | Land |
| Water | Both | Water | Water, fresh |
| Other | Both | None of the above. May include water quality parameters | Biological oxygen demand |

Flow classification is an extremely useful tool during flow mapping between LCA data sources, especially for large EF lists. By classifying the lists into smaller sub-groups, it is significantly easier to complete manual matching that may be necessary during a mapping process.

### 2.3. Source Data

To improve source interoperability, a systematic approach was taken to collect flowable names from federal emission, LCA and LCIA sources. Sources were expanded to make the list interoperable with other U.S. government-supported data sources. Including multiple types of sources in the creation of the FEDEFL improves the coverage of the EF list. This robust method of systematically building an EF list is seen as improving the ability of the FEDEFL to operate as a common source flow list. The following types of sources and sources were used in the creation of the FEDEFL.

#### 2.3.1. USEPA Sources

- TRI—Toxic Release Inventory. The TRI is a USEPA-produced data source for tracking potential toxic wastes and releases which includes available emission data for industrial facilities that are either from a specific industry, employ 10 or more full-time employees, or manufacture, process or otherwise use TRI-listed chemicals [17].
- NEI—National Emissions Inventory. The NEI is an estimate of criteria pollutants, criteria precursors, and hazardous air pollutants built using the Emissions Inventory System to collect and blend data from state, local, and tribal air agencies, and is updated every three years [18].
- RCRA—Resource Conservation and Recovery Act. RCRA is a USEPA reporting program to track non-hazardous solid waste and hazardous solid waste from 'cradle to grave' that requires large-quantity generators to report every two years [19].
- DMR—Discharge Monitoring Report. The National Pollution Discharge Elimination System (NPDES) collects reports for point source discharges to water bodies from permitted facilities [20]. DMR is a collection of the periodic (monthly, seasonally or semi-annual) water pollution reports derived from NPDES.
- eGRID—Emissions & Generation Resource Integrated Database. eGRID is a data source for air emissions and the generation of electrical power in the U.S. [21].
- PDP—the Pesticide Data Program. The USDA supports agricultural programs such as the PDP which monitors the residues of pesticides on agricultural products [22].
- Mineral Commodities Summary. The Mineral Commodities Summary is an annual report published by the U.S. Geological Survey covering the non-fuel mineral industry [23].
- Water Data for the Nation. The Water Data for the Nation provides water resource data about occurrence, quantity, quality, distribution, and movement of surface and underground waters [24].

#### 2.3.2. Database Sources

- GREET—The Greenhouse gases, Regulated Emissions, and Energy use in Technologies (GREET) Model created by Argonne National Laboratory [25]. GREET is a full LCA model supporting well to wheel, and fuel and vehicle cycles through material and disposal, covering energy and emissions for advanced and new transportation fuels.
- USLCI—U.S. Life Cycle Inventory (U.S. LCI) database by the National Renewable Energy Laboratory [26]. The U.S. LCI database is a collection of individual gate-to-gate, cradle-to-gate and cradle-to-grave life cycle assessments supporting material, component, or assembly within the U.S. The U.S. LCI is a repository for different North American stakeholders (e.g., industry associations) to provide public LCI data.
- Electricity Baseline LCI. The Federal LCA Commons Baseline [27] U.S. Regional Electricity LCI data available at https://www.lcacommons.gov/ (accessed on 4 June 2021).

#### 2.3.3. Software Sources

- openLCA—openLCA is a free open-source life cycle assessment software [28]. The openLCA EF list was utilized as an original source and the software has been used as a tool to support mappings. Training material has been developed to describe mapping to different EF lists, including the FEDEFL, within openLCA [29].

### 2.3.4. Life Cycle Impact Assessment Method Sources

The FEDEFL operates with common LCIA methods. The companion USEPA LCIA formatter tool (https://github.com/USEPA/LCIAformatter (accessed on 13 September 2021) was designed to use the FEDEFL as the EF structure for all LCIA and LCI methods processed through this package [30]. The FEDEFL and LCIA formatter process the following LCIA methods:

- TRACI—Tool for Reduction and Assessment of Chemicals and other environmental Impacts (TRACI). TRACI is a USEPA developed environmental impact assessment tool for characterization of life cycle data [31].
- ReCiPe—An impact assessment method for LCA first developed in 2008 through a joint effort from RIVM, Radboud University Nijmegen, Leiden University and PRé Sustainability [32].
- ImpactWorld+—A globally regionalized LCIA method based on a midpoint damage framework with four distinct complementary viewpoints [33].

### 2.3.5. Capturing Source Flows

The FEDEFL was designed to be an all-encompassing list capable of handling all flows to and from nature. The automated Standardized Emissions and Waste Inventories (StEWI) module developed by USEPA was used to generate comprehensive lists of flows used in USEPA datasets (i.e., inventory sources described above) [34]. Once chemicals were collected from these sources, chemicals were defined using two USEPA chemical databases, the Substance Registry Services (SRS) [35] and the Chemistry Dashboard [36]. SRS is the USEPA's authoritative resource on chemicals, biological organisms and other substances tracked and regulated by USEPA. The Chemistry Dashboard is a database for chemistry, toxicity, and exposure information for over 760,000 chemicals. The USEPA chemical databases were used to match chemical names, CAS No., and chemical formulas. SRS names are used in preference over chemistry dashboard naming. A common naming system for flows allows for the removal of duplicates and the correspondence of many sources of flowables and contexts to the FEDEFL.

### 2.3.6. Available Mappings

Mapping files are the essential tools of harmonization across platforms. Mapping files connect one source to another, usually at a flow level. Flow mappings are provided within FEDEFL software package for several commonly used existing LCI or LCIA sources to ensure compatibility with the FEDEFL and each other. In many cases, mapping files are generated by matching flowables and contexts separately. This reduces the potential of inconsistent matching across flowables when tracked in multiple contexts and inconsistent contexts. Upon generating a mapping file, flow Universal Unique Identifiers (UUID) are checked to ensure that the target flowable and context exists in the FEDEFL. The methods section describes the simplistic mapping method used to determine interoperability for the analysis.

There are currently several available mapping files. Existing mapping files were used to support the analysis for this paper and guide the selection of the sources for the analysis. Currently, mapping between the FEDEFL and the GREET model, USLCI database version 2019Q4, and TRACI 2.1 are available. The open-source software openLCA has been working with the Federal LCA Commons to integrate flow list management changes; therefore, a mapping to the reference list from openLCA is also available [29]. Since openLCA is a repository for various datasets, original data sources connected with the openLCA reference list were also not a part of the mapping process. In addition to Federal LCA Commons sources, mappings to LCIA methods ReCiPe and ImpactWorld+, and LCA sources IDEA and ecoinvent, are available. All these mappings are one-way mappings to the FEDEFL and are available online on the FEDEFL github repository. One-way mappings are mappings that have been completed as a part of the Federal LCA Commons and have not in the

instances of ReCiPe, ImpactWorld+, IDEA or ecoinvent been reviewed or approved by the original source.

## 3. Materials and Methods

Three analyses were conducted on the FEDEFL version 1.0.8 to test interoperability improvements. The analyses employed the same methods to match EFs or components of these flows as described in previous studies [10,13]. The analyses used publicly available sources that were available to the authors. The first analysis was conducted to determine how capable the FEDEFL is at mapping to other LCA sources. Two databases and datasets (ecoinvent and USLCI), one software (openLCA), and two LCIA methods (TRACI and ImpactWorld+) with existing mappings to FEDEFL were selected for analysis. The number of unique flowables that were mapped were compared with the total number of unique flowables in each of the sources. In this analysis, unique flowables refers to the name of the flows, excluding any context or unit information.

The second analysis was performed to determine the current interoperability between other LCA EF lists. The sources of EF lists is limited to either publicly available EF lists or EF lists that were available through previous and ongoing collaborative efforts, such as the critical review [10,13]. The authors acknowledge that additional databases and datasets, software and LCIA methods exist, but were not included in the comparison. This comparison is not meant to be an exhaustive comparison of all LCA sources, but a representation of general issues between LCA sources. Flows from each list were compared to flows in every other flow list using name-to-name matching. For the purposes of simplicity, context information was not included when matching flows, and only unique flows (i.e., flowables) were used from each list. Ten sources were selected for this analysis: four databases and datasets, three pieces of software and three LCIA methods. Table 3 provides the ten source names and versions that were used for this analysis

**Table 3.** Flow list analysis sources.

| Source Type | Source Name | Version |
|---|---|---|
| Databases and datasets | USEEIO | v1.1 [37] |
| Databases and datasets | USLCI | Q4v2019 [26] |
| Databases and datasets | IDEA | v2.3 [38] |
| Databases and datasets | ecoinvent | v3.6 [39] |
| Software | openLCA | Reference flow list (2020) [28] |
| Software | SimaPro | Used in 2018 GLAD critical review [10] |
| Software | GaBi | Used in 2018 GLAD critical review [10] |
| LCIA methods | TRACI | v2.1 [31] |
| LCIA methods | ReCiPe | 2016 release [32] |
| LCIA methods | ImpactWorld+ | [33] |

Lastly, an analysis was conducted to use the FEDEFL as an intermediate flow list in mapping LCI databases to LCIA methods. Flow lists from two LCI databases—USLCI and ecoinvent—and lists from three LCIA methods were used in the analysis. The versions of the LCI and LCIA sources used are listed in Table 3. For this analysis, the source lists were both mapped to FEDEFL and then compared to see if there was an improvement in matching capabilities between the original source to source matching and the matching using the FEDEFL as a common EF list.

## 4. Results

Table 4 shows how many flows from each of the five sources were capable of being mapped to the FEDEFL from the total flows within the source. Mappings were possible for 85-98% of flows from these five other flow lists. Most of the flows that remained unmappable either contained information that was not possible to capture with the FEDEFL, such as long-term emissions in ecoinvent, or flows that were unidentifiable, such as chemicals that lacked CAS numbers.

**Table 4.** Available mappings analysis.

|  | Ecoinvent | USLCI | OpenLCA | TRACI | ImpactWorld+ |
|---|---|---|---|---|---|
| Mapped | 3751 | 4351 | 4605 | 3568 | 3457 |
| Total | 4323 | 4450 | 5216 | 3950 | 4048 |
| % of flows mapped | 87% | 98% | 88% | 90% | 85% |

In Table 5, an analysis comparing flow names from different sources shows some of the challenges associated with interoperability. The table in Table 5 is read as the percentage of flowables found in the source list in the first column that can be found in the source list in the top row. For example, 55% of flowables from ecoinvent can be found in USLCI. Lists were viewed as having significant matching capabilities if greater than 50% of unique flowables were found to match. Lists were defined as having less than marginal matching capabilities if less than 25% of unique flowables matched. Based on name–name matching between sources, 25% of dataset comparisons had significant matches, while 56% of comparisons showed less than marginal matches.

An example of significant flow matching capabilities can be found for USEEIO flowables found in openLCA, as 64% of flowables from USEEIO matched with a flowable in openLCA. IDEA was found to have marginal matching capabilities with openLCA with 31% of flowables matching. In fact, nine sources found marginal matching capabilities with openLCA and seven out of nine sources found significant matching capabilities with openLCA. openLCA had the greatest capacity for automated name–name matching (defined as the greatest number of significant and marginal matching capabilities based on openLCA column). It is to be expected that when other sources are matched with software sources, higher matching capabilities will be observed. This is because software EF lists tend to be created as an amalgamation of EFs from other sources and are more inclusive in their structure.

**Table 5.** Comparison of interoperability between LCA sources.

| Legend | |
|---|---|
| | ≤25% |
| | ≥50% |

| | | Databases and Datasets | | | | Software | | | LCIA Methods | | |
|---|---|---|---|---|---|---|---|---|---|---|---|
| | | USEEIO [1] | USLCI [2] | IDEA [3] | Ecoinvent [4] | OpenLCA | SimaPro [5] | GaBi [5] | TRACI [6] | ReCiPe [7] | ImpactWorld |
| **Databases and Datasets** | USEEIO [1] | | 30% | 14% | 34% | 64% | 51% | 11% | 44% | 41% | 44% |
| | USLCI [2] | 23% | | 7% | 18% | 74% | 54% | 17% | 53% | 18% | 32% |
| | IDEA [3] | 27% | 17% | | 16% | 31% | 27% | 14% | 25% | 26% | 25% |
| | Ecoinvent [4] | 5% | 55% | 1% | | 69% | 11% | 2% | 26% | 25% | 39% |
| **Software** | OpenLCA | 9% | 13% | 2% | 12% | | 59% | 4% | 32% | 28% | 42% |
| | SimaPro [5] | 8% | 11% | 2% | 12% | 66% | | 3% | 34% | 30% | 44% |
| | GaBi [5] | 19% | 37% | 13% | 25% | 45% | 34% | | 17% | 19% | 27% |
| **LCIA Methods** | TRACI [6] | 6% | 7% | 2% | 10% | 71% | 30% | 1% | | 75% | 63% |
| | ReCiPe [7] | 15% | 9% | 6% | 12% | 73% | 69% | 5% | 86% | | 71% |
| | ImpactWorld | 12% | 13% | 4% | 16% | 95% | 79% | 4% | 64% | 62% | |

[1] USEEIOv1.1; [2] USLCIQ4v2019; [3] IDEA 2.3; [4] ecoinvent 3.6; [5] SimaPro and GaBi from the 2018 GLAD critical review; [6] TRACI 2.1; [7] ReCiPe 2016.

Databases and datasets tend to be narrower in scope and are, therefore, most likely to exhibit the lowest matching rates, as seen in the first four columns of Table 5 (under the databases and datasets header). However, when databases and datasets are matched with software sources, the larger, more inclusive nomenclatures of the software tend to allow for higher matching rates. The low number of significant matches found between databases and datasets and LCIA methods (found in the last three columns of the table) shows why it is important for mapping files to exist between sources. A lack of connectivity between databases, datasets, and LCIA methods can lead to inaccurate LCA results. Therefore, most software sources have a mapping file with LCIA methods. However, mappings can be time consuming to produce and need to be consistently updated for interoperability between sources to be maintained.

When the FEDEFL is used as an intermediate mapping, more flows from USLCI and ecoinvent Figure 1 map to LCIA methods. The matching percentage between LCI databases and LCIA methods increased 38% to 277% across the different LCI and LCIA sources.

The variation in improvement in matching can be attributed in large part to the framework and structure of the original sources. Some of the databases and LCIA methods have similar naming structures or nomenclature rules when compared with other sources. This could account for minimal increases in interoperability in select cases when using a common standard list. Additionally, it is expected that not all flows should match between an LCI source and LCIA method, as there are some flows that will not result in an impact for the categories assessed in the selected LCIA method. By adhering to the strict definition of an EF used by the FEDEFL, not all the original source EFs in USLCI are considered EFs. For example, 'volume occupied, reservoir' or 'process effluent' from the USLCI are not mapped to the FEDEFL since the FEDEFL does not consider space occupation an EF and process effluent is also not a specific substance, but rather represents a mix of unknown substances. In the case of 'volume occupied, reservoir', this also does not correspond to impact assessment categories in the covered LCIA methods, so exclusion is not expected to affect the robustness of study results. In the case of 'process effluent,' the original LCI developer did not provide sufficient information on the composition of the flow to allow for incorporation in an LCIA method. While tools such as the FEDEFL improve the interoperability and structure of LCA models, model results remain dependent on the quality of the underlying source data.

As the GLAD critical review and Table 5 revealed, there currently exists a gap between LCA sources' EF structures and LCIA methods' EF development [10]. The use of a standard list, such as the FEDEFL can be a vital step forward to closing the gap between LCIA method development and LCI databases.

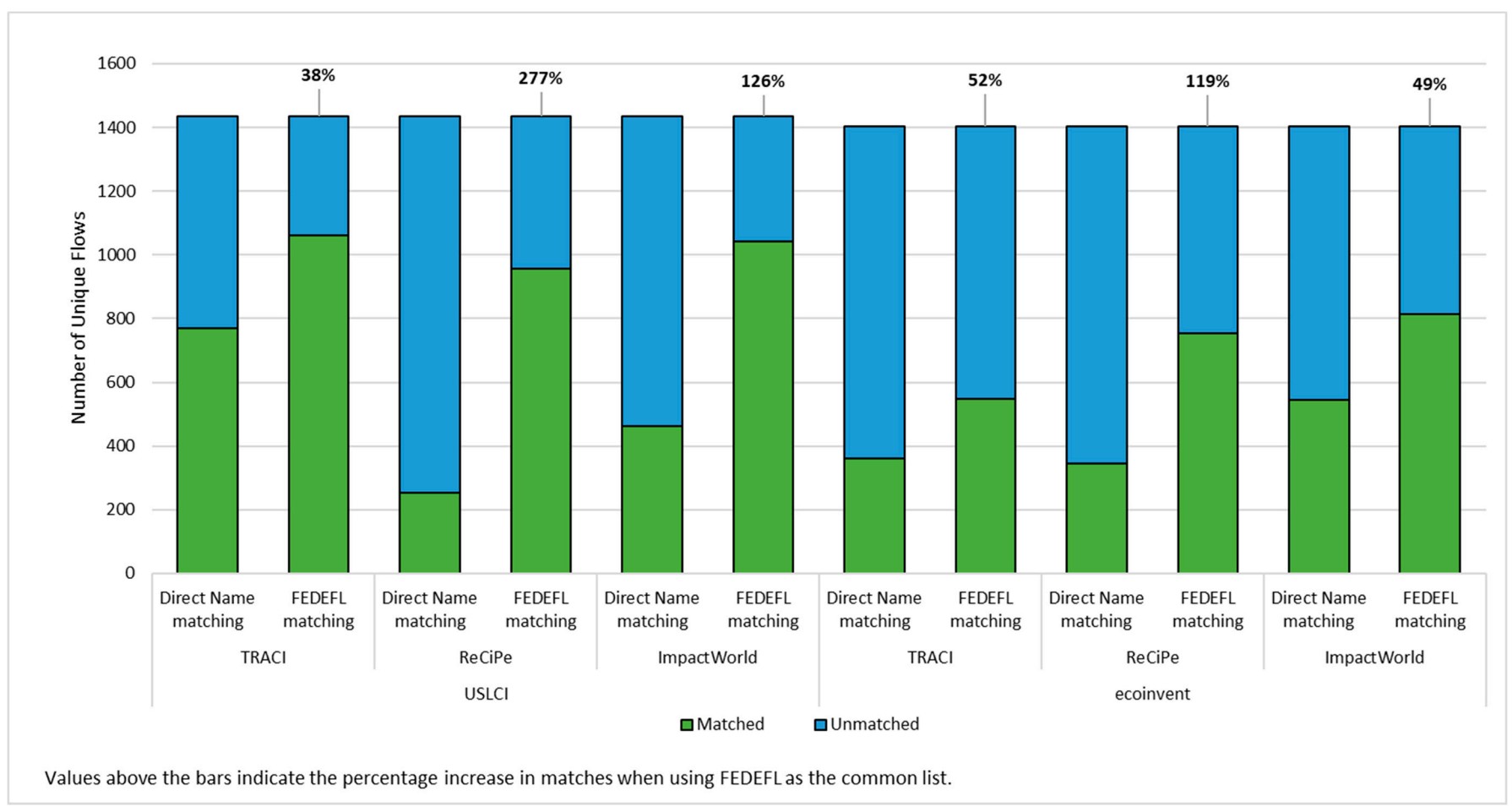

**Figure 1.** Analysis of improved matching using FEDEFL as the common list.

## 5. Discussion

Due to the diverse nomenclatures and LCA development methodologies that are currently available, users face a significant challenge in synthesizing and harmonizing data from different sources within the LCA community. As shown in Table 5 significant differences in nomenclature between sources still exist, as found by Edelen et al. [10]. This lack of EF connectivity can lead to flows inadvertently being excluded from an LCIA analysis, which presents significant challenges to LCA developers and can lead to erroneous results. In the past, the only alternative for developers is to spend significant resources to manually map EFs for each individual LCA. With the automated tools in the FEDEFL toolbox, users can greatly decrease the resources necessary to harmonize EFs and improve the accuracy of LCA study results.

Users should also be aware of the limitations of mapping files. The mapping files available through the FEDEFL repository are one way mapping files; mappings are completed by FEDEFL to map from other data sources to the FEDEFL. While a flow may be mappable in one direction, it does not mean that the reverse mapping is accurate. In mapping files, flows are mapped using different mapping conditions, "=", "~", ">", and "<". While some flows are ideal matches and can be matched using the "=" signs, this is not always true for all mappings. For example, in one source you may see the flowable name "Silver(II)" and in another list they only have "Silver". The "Silver" flowable is not an exact match to "Silver(II)" and can include more silver ions that just "Silver(II);" therefore, a "<" with the greater side directed at "Silver" would be used. This type of mapping is performed at the discretion of the mapping creators and often is performed in instances where the chemical has a known impact. However, not all flows are mapped to proxy flows and users should be careful when using mappings as some flows with known impacts may disappear during the mapping process or end up mapped to flows that the user does not find to be an acceptable proxy flow.

The mapping files posted through the FEDEFL were created by developers of the FEDEFL and not necessarily with the collaboration of the original data sources. This creates limitations as flows from the original sources may have been unclear or misinterpreted by the FEDEFL developers due to lack of metadata. Examples of developer-dependent decisions include mapping flows with errors, such as misspelled flows. Often developers are left to either leave misspelled flows out of mappings or map to the flow they believe the original data source was referencing. Mapping developers must make decisions such as whether to map flows with greater specificity to flows with lesser specificity. This can occur for both context information and for flowable information. An example of this challenge is particulate matter. Some databases have flows such as "Particles (PM2.5)", "Particles (PM0.2–PM2.5)", and "Particles (PM0.2)" where the FEDEFL only has "Particulate matter, $\leq$2.5 µm". An example where context information may be more specific is the inclusion of some databases of flows with "long-term" context subcompartment and the exclusion from the FEDEFL. Decisions such as these create significant challenges for projects such as that undertaken by the GLAD nomenclature working group to provide dual direction mapping files from four LCA developers (i.e., ecoinvent, ILCD, FEDEFL and IDEA), as different developers must spend significant time agreeing on these decisions. The GLAD nomenclature working group has provided a report of the challenges faced during this project in addition to the mapping files so that users may better understand the limitations of mappings.

The removal of exchange information from flow names does have an impact on the mapping abilities of FEDEFL flows with other EF lists and ultimately creates challenges with LCA studies. The inclusion of exchange information in a flow name, such as 'biogenic' or the flow location, allows software to treat these flows as separate and, therefore, trace impacts separately. However, with the removal of this information these flows are no longer tracked within a LCA system and a LCA developer must manually track these flows to include in an impact assessment. Including such exchange information in a flow name can, however, lead to long and complicated flow lists. For example, many water flows

include location name to be able to track regional water scarcity. Future improvements of the different LCA software tools could instead link the location to the exchange rather than directly to the flow name. Currently, an alternative to manual tracking of these flows does not exist.

## 6. Conclusions

The future efficacy of LCA as a decision support tool requires investments to improve interoperability. However, there are limitations to interoperability if the LCA community continues to lack consensus on adopting a single nomenclature. Mapping files will have to be consistently produced and updated as source lists evolve, and new versions are released. Manual mapping of certain flows will continue to play a critical and time-consuming role in interoperability. A lack of consistent structure and nomenclature around flow context information also adds complexity to the interoperability of EF nomenclature.

The development of the FEDEFL framework and the resulting flow list was demonstrated to improve interoperability. The structure of the FEDEFL nomenclature has been used as a roadmap for other LCA databases and data developers to improve the interoperability of lists through the increased use of metadata to identify flows and through the utilization of flow components to simplify the flowable names. The novel FEDEFL framework is adaptable and has been utilized in the development of the GLAD nomenclature mapping files [13]. This effort by the Federal LCA Commons to develop the FEDEFL toolbox has also improved the ability of management of nomenclatures and mapping files and increased access to such resources through the publicly available github repositories and version control of lists.

**Author Contributions:** Conceptualization: W.W.I. and A.N.E. Software: B.Y., W.W.I. and A.N.E. Data Curation: A.N.E., B.Y., S.C. and W.W.I. Investigation: A.N.E., B.Y., S.C. and W.W.I. Methodology: A.N.E., S.C., B.Y. and W.W.I. Supervision: W.W.I. and S.C. Validation: A.N.E., B.Y. and W.W.I. Visualization: A.N.E. and S.C. Writing: A.N.E., W.W.I., B.Y. and S.C. All authors have read and agreed to the published version of the manuscript.

**Funding:** This research was funded by the USEPA's Sustainable and Healthy Communities Research Program. This research was supported through USEPA contract EP-C-16-015, Task Order 68HERC19F0292 with Eastern Research Group (ERG).

**Institutional Review Board Statement:** The U.S. Environmental Protection Agency, through its Office of Research and Development, funded and conducted the research described herein under an approved Quality Assurance Project Plan (K-LRTD-0030017-QP-1-3). It has been subjected to the Agency's peer and administrative review and has been approved for publication as an EPA document. Mention of trade names or commercial products does not constitute endorsement or recommendation for use.

**Data Availability Statement:** Supporting analyses can be found at http://doi.org/10.23719/1527971.

**Conflicts of Interest:** The authors have no conflict of interest.

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
