# Peer review of "Life Cycle Data Interoperability Improvements through Implementation of the Federal LCA Commons Elementary Flow List"

_applsci, doi:10.3390/app12199687_

Round 1

Reviewer 1 Report (New Reviewer)

This study addressed an uncertainty associated with LCA results from mapping elementary flows. The authors provided a comprehensive background review of the federal elementary flow list (FEDEFL), as well as the novel framework aiming to improve elementary flows’ interoperability in LCA modeling. They demonstrated that implementing the FEDEFL framework could noticeably improve the mapping capabilities and thus increase the level of interoperability.

Author Response

Reviewer 1

This study addressed an uncertainty associated with LCA results from mapping elementary flows. The authors provided a comprehensive background review of the federal elementary flow list (FEDEFL), as well as the novel framework aiming to improve elementary flows’ interoperability in LCA modeling. They demonstrated that implementing the FEDEFL framework could noticeably improve the mapping capabilities and thus increase the level of interoperability.

Thank you for your feedback

Reviewer 2 Report (New Reviewer)

The paper presents a novel framework and structure for elementary flows, which is implemented in the Federal Elementary Flow List. The interoperability of ten EF lists, two LCA databases, three life cycle impact assessment (LCIA) methods, and five LCA software sources is assessed with and without use of the FEDEFL as an intermediary in flow mapping. 

The paper is overall well written. The state of the art of the federal elementary flow list is introduced detailed. Three analyses were conducted on Version 1.0.8 of the FEDEFL to test interoperability improvements.

Author Response

Reviewer 2

The paper presents a novel framework and structure for elementary flows, which is implemented in the Federal Elementary Flow List. The interoperability of ten EF lists, two LCA databases, three life cycle impact assessment (LCIA) methods, and five LCA software sources is assessed with and without use of the FEDEFL as an intermediary in flow mapping. The paper is overall well written. The state of the art of the federal elementary flow list is introduced detailed. Three analyses were conducted on Version 1.0.8 of the FEDEFL to test interoperability improvements.

Thank you for your feedback

Reviewer 3 Report (New Reviewer)

The author (s) developed a framework to increase the interoperability of LCA. This manuscript adds considerable value to the existing knowledge base. However, the following points may be taken into account to uplift the quality of the manuscript.

1. Abstract – Note that a good abstract should contain five main elements: introduction, problem statement, methodology, contributions, and results.  Authors should rewrite the abstract according to this. Do not use acronyms in your abstract, keywords, and title, include a sentence about your findings, discussions, and conclusions in your abstract and underscore the scientific value added by your paper.

2. The introduction section should highlight the need for the study and its contribution to the existing knowledge base. Also, it should introduce the methodology adopted. The introduction should clearly state the research questions and targets first. Then answer several questions: Why is the topic important (or why do you study it)? What are the research questions? What has been studied? What are your contributions? Why is it to propose this particular method? An outline of the paper must be included. The introduction comprises only 7 lines and doesn't clearly indicate anything. 

3. subsection 1.2 may be shifted to Introduction (Section 1)

4. The major defect of this study is the debate or Argument is not clearly stated in the introduction session. Hence, the contribution is weak in this manuscript. I would suggest the author(s) enhance the theoretical discussion and arrive at the debate or argument.

5. Research gaps may be included after the background of the study.

6. Detailed justification has to be provided for choosing the methodology. 

7. The author(s) need to emphasize more the limitations, future scope, and research implications of the study. 

Best wishes..!

Author Response

Reviewer 3

  1. Abstract - Note that a good abstract should contain five main elements: introduction, problem statement, methodology, contributions, and results. Authors should rewrite the abstract according to this. Do not use acronyms in your abstract, keywords, and title, include a sentence about your findings, discussions, and conclusions in your abstract and underscore the scientific value added by your paper.

We agree. Most of the abstract is already in this form, but we made some additional changes based on your other comments to improve it. We also removed use of all acronyms. Here is our annotation to describe how sentences in the revised Abstract fit the five element structure that you describe:

Introduction: As a fundamental component of data for life cycle assessment models, elementary flows have been demonstrated to be a key requirement of LCA data interoperability. However, existing elementary flow lists have been found to lack sufficient structure to enable improved interoperability.

Problem Statement: The Federal LCA Commons Elementary Flow List provides a novel framework and structure for elementary flows, but the actual improvement this list provides to interoperability of LCA data has not been tested.

Methodology: The interoperability of ten elementary flow lists, two LCA databases, three life cycle impact assessment methods, and five LCA software sources is assessed with and without use of the Federal LCA Commons Elementary Flow List as an intermediary in flow mapping.

Results: This analysis showed that only approximately 25% of comparisons between these sources resulted in greater than 50% of flows being capable of automatic name-to-name matching between lists. This indicates that there is a low level of interoperability when using sources with their original elementary flow nomenclature, and elementary flow mapping is required to use these sources in combination. Mapping capabilities of the Federal LCA Commons Elementary Flow List to sources were reviewed and revealed a notable increase in name-to-name matches.

Contribution: Overall, this novel framework is found to increase LCA interoperability.

  1. The introduction section should highlight the need for the study and its contribution to the existing knowledge base. Also, it should introduce the methodology adopted. The introduction should clearly state the research questions and targets first. Then answer several questions: Why is the topic important (or why do you study it)? What are the research questions? What has been studied? What are your contributions? Why is it to propose this particular method? An outline of the paper must be included. The introduction comprises only 7 lines and doesn’t clearly indicate anything.

We removed section 1.2 and integrated it into the Introduction. We redrafted the introduction to clearly present the well known challenges of interoperability and a brief history of what led to the creation of the FEDEFL framework as a solution. We clarified that the objective was to complete an analysis of whether the new FEDEFL framework improved interoperability and in what ways the interoperability was improved. The introduction also introduces the history of using similar analysis to review and analyze interoperability issues.

  1. subsection 1.2 may be shifted to Introduction (Section 1)

We removed the section title and integrated the text into the Introduction.

  1. The major defect of this study is the debate or Argument is not clearly stated in the introduction session. Hence, the contribution is weak in this manuscript. I would suggest the author(s) enhance the theoretical discussion and arrive at the debate or argument.

We have addressed this defect through our revisions to the Abstract and Introduction based on your earlier recommendations (1-3).

  1. Research gaps may be included after the background of the study.

We have added some additional literature references and described these gaps to the Introduction.

  1. Detailed justification has to be provided for choosing the methodology.

We added a justification to the choice of the methodology in the Materials and Methods section by referring to previous research that has used the same methods to analyze interoperability issues.

  1. The author(s) need to emphasize more the limitations, future scope, and research implications of the study.

We agree and have done this through revisions made in the Abstract, Introduction and Conclusion. In the abstract and introduction, we clarify that the main research implication of the analysis is to test interoperability improvements through development of the FEDEFL (see response to comment #2). In the conclusion, we added a discussion about the remaining limitations such as the lack of consensus in the LCA community around adopting a single nomenclature. In the conclusion, we now also describe other limitations such as the continual resources required to maintain EF lists and associated mapping files. Finally, in the Conclusion we reiterate the research implication of this paper of improved interoperability in LCA data sources through use of a structured standardized list.

This manuscript is a resubmission of an earlier submission. The following is a list of the peer review reports and author responses from that submission.

Round 1

Author Response

  1. Section: 1 Introduction

On page 1, Line 30-36, review or add the LCA standard ISO14040 and ISO14044 (2006).

The authors mention the contributions of the ISO14040 and 14048 series, which specifically address nomenclature in lines 51-53.

“The International Organization for Standardization (ISO) has also provided support for LCA interoperability through the ISO 14000 series. Specifically, the ISO 14048 standard provides LCA guidance on nomenclature [4].”

  1. Lack of review LCA data in Asia such as LIME method (Japan)

The authors were unaware of LIME at the time of the analysis. After doing a search the authors were unable to locate a publicly available EF list for LIME and have opted instead to include this text in the Material and Methods section:

"The sources of flow lists are limited to either publicly available EF lists or EF lists that were available through previous and ongoing collaborative efforts. The authors acknowledge that additional databases and datasets, software, and LCIA methods exist, but were not included in the comparison. This comparison is not meant to be an exhaustive comparison of all LCA sources. But a sample representation of general issues between LCA sources."

Reviewer 2 Report

Dear authors, your manuscript is addressing an important topic at the moment. The interoperability and the data discrepancies in using different LCA databases is really  a hot topic.

I am very familiar with this topic but it was hard to understand everything in your manuscript. I believe that the "standard" reader cannot follow. The analysis and the method itself is correct, but you have to search for bits and pieces all over the paper. In addition, you have submitted the paper to the "Computing and AI" section. The relevance to computing is relative poor here. My suggestion is to restructure the paper.

First, the Introduction is far too long. Please shorten it to a short introduction about the problem itself and the motivation of the paper. It is already in this manuscript but not in the beginning.

The part from line 88ff should be a new chapter 2 with the state of the art. Here you already mix it up the the "Material and Methods" section. Please separate the State of the Art part from the Methods part. As the FEDEFL is VERY intensively describes, you should think about a section as "supplementary information". I thought that some parte were repeated in too much detail.

In addition FEDEFL and GLAd should be better diffentiated. They are separate things and it is hard to understand by reading your paper.

Also, the section "software" is a standalone section. Is this the software used in your assessment? Then is definitely belongs to section method. In addtion, the version 1.0.8 of FEDEFL belongs to the method section. In some parts line 274 ff to line 350ff seems to be more a product advertisement than a scientific paper. This needs to be shortened or placed into the "Supplementary" section.

LCIA is also a standard in LCA. Here again, the reader might think that this is a special tool of FEDEFL.

Section 1.3 was the part I was missing early in the introduction.

Chapter 2 is too short. Parts of the intro definitely need to be shifted here. For me it is still not clear what you have actually done. Maybe a flow sheet would here? You just performed an analysis with FEDEFL? Isn't FEDEFL intended to serve just as that? The explanation is also very unspecific as: "the number of unique flows" (what is that?), "excluding the context" (what context?), "other LCA flow lists", "Ten sources" (without names). It was also not explained why these sources were selected.

Chapter 2 needs to be rewritten.

Chapter Discussion is fine when the Method section is explained in detail. The information received might be useful for the community.

Chapter Conclusion should not contain abbreviations. The conclusion is also rather a product advertisement than a scientific conclusion.

But more or less, it is rather useful for the LCA community. I still do not understand the submission section. Why is this submitted to the Computing and AI section? Only a software was applied and there is no computing approach .

Even the manuscript is useful (for the LCA community) I think that computer scientists are not interested.

Author Response

The authors' responses are in red. Reviewers' comments are in black.

Reviewer 3 Report

Dear Authors,

the article discusses an important topic in the field of LCA with which LCA practitioners regularly deals with. The article briefly introduces the structure of FEDEFL and compares the nomenclature in selected databases.

I consider the article sufficient, but I also have comments and questions:

 - Line 232 – the first sentence is probably missing some words

 - During the analysis, was the choice of the mapped and compared flows random? Which flows were mapped?

 - Figure 3 – I find it difficult to understand the graphics, maybe you should add some more description.

Best,

Author Response

Reviewer comments in black

Authors comments in red

Round 2

Reviewer 2 Report

Dear authors,

I still do not see a progress here and most of my remarks are still open.